# The Use of Rhizospheric Microorganisms of *Crotalaria* for the Determination of Toxicity and Phytoremediation to Certain Petroleum Compounds

**DOI:** 10.3390/plants15010103

**Published:** 2025-12-29

**Authors:** Ana Guadalupe Ramírez-May, María del Carmen Rivera-Cruz, María Remedios Mendoza-López, Rocío Guadalupe Acosta-Pech, Antonio Trujillo-Narcía, Consuelo Bautista-Muñoz

**Affiliations:** 1Knowledge Generation Line 1 (KGL1), Management and Conservation of Natural Resources, Doctorate in Agricultural Sciences in the Tropics, College of Postgraduates Tabasco Campus, Cardenas 86500, Mexico; ramirez.ana@colpos.mx; 2Agricultural and Environmental Microbiology Laboratory, Collage of Postgraduates Tabasco Campus, Cardenas 86500, Mexico; 3Institute of Applied Chemistry, Veracruzana University, Xalapa 91190, Mexico; remendoza@uv.mx; 4Knowledge Generation Line 2 (KGL2), Sustainable Agricultural and Livestock Production Systems, College of Postgraduates Tabasco Campus, Cardenas 86500, Mexico; acosta.rocio@colpos.mx (R.G.A.-P.); cbautistam@colpos.mx (C.B.-M.); 5Academic Group of Energy and Environment, Popular University of Chontalpa, Cardenas 86556, Mexico; atrujillonarcia@gmail.com

**Keywords:** arbuscular mycorrhizal fungi, Rhizobia, *Crotalaria pallida*, linear alkanes (C12 to C26)

## Abstract

Microbial toxicity tests in the rhizosphere play an important role in the risk assessment and phytoremediation of chemical compounds in the environment. Tests for the inhibition of nodule number (NN), Rhizobia in the rhizosphere (RhR), *Rhizobium* in nodules (RhN) and arbuscular mycorrhizal fungi (AMFs) are important to evaluate the toxicity as well as the removal of total petroleum hydrocarbons (TPHs), 15 linear alkanes (LAs), and total linear alkanes (TLAs). The inhibition and removal was evaluated at 60 (vegetative stage, VS) and 154 days (reproductive stage, RS) of the life cycle of *Crotalaria incana* and *Crotalaria pallida* in soil with four doses of CRO (3, 15, 30, and 45 g/kg) plus a control (16 treatments). Results indicated that RhN and five structures of the AMFs present an index of toxicity (IT < 1), and the microbiological variable is inhibited by the CRO. RhR exhibits a hormesis index (IT > 1) that is stimulated by the CRO in the VS and RS for *C. incana* and *C. pallida*. The highest removal of TPHs (77%) was in the rhizosphere of *C. incana* in the RS with 45 g/kg of CRO. *C*. *pallida* removed the greatest amount of TLA (91%). There was a positive correlation between the RhR and the removal of TPHs, TLA, and LAs (higher molecular weight). It could be argued that symbiotic microorganisms are significant for use in toxicity testing, and the rhizosphere of *C. incana* and *C. pallida* can be used for the phytoremediation of HTPs and ALs in loamy-clay soil contaminated with CRO.

## 1. Introduction

Crude oil (CRO) is a complex mixture that contains hydrocarbons [saturated (linear n-alkanes, branched alkanes), polycyclic aromatics, resins, and asphaltenes] [1] and heavy metals (Zn, Cu, Pb, Ni, Mn, Co, and 27 other) [2]. Total petroleum hydrocarbons (TPHs) comprise the light, medium, and heavy fractions of hydrocarbons in the CRO mixture. C12 to C26 linear n-alkanes (LAs) constitute the liquid and solid fractions of TPHs. TPHs and LAs have toxic, carcinogenic, and mutagenic potential for living beings [3]. TPHs and LAs continue to accumulate after the accidental contamination of soil covered with tropical legumes, particularly of the genus *Crotalaria*, which contains 702 species and whose center of diversity is tropical Africa, Madagascar, and other parts of the world [4].

*C. incana* and *C. pallida*, found in the Mexican humid tropics, provide a food source for humans and animals and a source of protein in the absence of animal protein, and these species are used in crop rotation to fix N [5]. The roots of *Crotalaria* establish a mutualistic relationship with Rhizobia and AMFs. When exposed to hydrocarbons, Rhizobia and AMFs are either growth-inhibited or increase in population via oxidative stress, which alters the structure and function of cells [6].

Microbial toxicity tests play an important role in various scientific and technical fields, including the risk assessment of chemical compounds in the environment. The use of toxicity tests that study detrimental changes in the function, appearance, and growth of microorganisms exposed to chemical substances, and have been standardized by the ISO and OECD, is becoming more commonplace [7]. There are publications related to the inhibition of microorganisms in the rhizosphere of plants exposed to CRO, and they report that the toxicity index is obtained from the dose–response relationship of the microbiological variable [8,9]. The indicator hormesis is a positive stimulus of the plant and microorganisms to low doses of contaminant; the plant and microorganisms exhibit a process of acclimatization and adaptation to stress due to abiotic factors [10]. Hormesis plays a crucial role in the control and remediation of soil contamination [11].

The mutualistic microorganisms and nodules in the root of *Crotalaria* have rarely been studied as an indicator of the toxicity and phytoremediation of soils contaminated with CRO. The Rhizobiaceae that live are heterotrophic and fix N only when hosted in legumes root nodules [5]; in addition, legumes also benefit from arbuscular mycorrhizal fungi (AMFs), which supplement low-mobility nutrients in the soil, particularly P [12]. Free-living rhizobia and AMFs can play dual roles as biological indicators of stress and can remove TPHs and LAs from the soil. In this context, ref. [13] under conditions of biotic and abiotic stress, owing to the complexity in the exchange of signals and nutrients, the growth and development of microbes can be inhibited, allowing symbiosis with the root system.

The rhizobia in rhizosphere (RhR) and AMFs associated with plants intervene individually or through cometabolism in the degradation of petroleum hydrocarbons and facilitate the restoration of ecosystems damaged by industrial activity [14,15]. The ability of rhizospheric microorganisms to degrade LAs of up to 44 carbons (C44) present in CRO has been observed [16]. The AMFs associated with *Avena sativa* induces up to 60% removal [17], and rhizobia in the rhizosphere (RhR) can use petroleum hydrocarbons as a source of carbon and energy for its metabolism before invasion of the root system [18]; however, potential in the degradation of hydrocarbons in the soil for RhN has not been demonstrated, although the presence of petroleum hydrocarbons in the rhizosphere has been shown to inhibit viability [19].

It has been shown that the age of a plant in contaminated soil is a factor that induces a rhizospheric system with genetic signals and differentiated organic and inorganic root exudates to be colonized by endophytic microorganisms to form intraradical structures and participate in plant nutrition and the rhizoremediation of contaminants [20]. Genetic signals induce mutualistic symbioses that depend on the nutritional requirements of the plant on the basis of plant age and stress factors such as contaminants in the soil [21].

In the reproductive stages (RS) of the plant life cycle, the activity of the roots and the absorption of nutrients generally decrease, mainly as a result of the decrease in the supply of carbohydrates to the roots, although some plants require N absorption post-reproduction [12]; however, the remobilization of nutrients is particularly important during RS, when the seed, fruits, and storage organs are formed [22]. The plant in each growth stage exudes organic and inorganic molecules through the root system that function as signals for mutualistic association with beneficial endophytic microorganisms that form nodules and intraradical structures to transport N and P [23] and induce TPHs and LAs degradation in the rhizosphere.

Background information indicates that phytoremediation is an inexpensive and long-term technology that uses pollutant-tolerant green plants and their rhizospheres, which host microbial symbionts that serve dual purposes, to supplement nutrients and to use petroleum hydrocarbons for their metabolism by degrading or stabilizing them [24,25]. The use of the legumes rhizosphere in phytoremediation technologies has been demonstrated for soils with low doses of petroleum hydrocarbons [26].

Legumes that grow naturally in soils contaminated with CRO near petroleum infrastructures in southeastern Mexico have been little studied in toxicity tests to select Crotalaria and rhizosphere-associated microbiological parameters that support their potential as indicators of toxicity and their relationship with the phytoremediation of CRO-derived hydrocarbons. On the basis of the above, the objectives of this research were (a) to evaluate the effects of CRO on the population of free-living RhR, the number of nodules, the population of RhN, and the AMFs associated with *C. incana* and *C. pallida* in the vegetative and reproductive stages; (b) to determine the effects of CRO on microbiological variables of mutualistic association that support the use of systems testing the toxicity; and (c) to determine the potential of rhizobia and AMFs associated with the rhizosphere of *C. incana* and *C. pallida* for use in phytoremediation applications for the recovery of clay loam soils contaminated with TPHs y ALs (C12–C26) derived from oil in the Mexican humid tropics.

## 2. Results

### 2.1. Effect of CRO on Rhizobia and Nodules at Different Stages of the Crotalaria Life Cycle

The rhizobia in rhizosphere (RhR) population, the number of nodules (NN), and the rhizobium in nodule (RhN) population, as well as the percentage of hyphae, arbuscules, vesicles, spores, total root colonization, and rhizosphere spores of AMFs, showed statistically significant differences due to the CRO dose during the growth (VS) and flowering-fruiting (RS) stages in both Crotalaria legume species (Table 1). The highest rhizobia population in the rhizosphere was 1792 × 10^4^ CFUs g^−1^, found in soil with 45 g/kg of CRO planted with *C. pallida* in the RS, and the lowest (8 × 10^4^ CFU) was found in *C. pallida* in the VS in established plants in soil without CRO (Table 1). The effect of plant species on the RhR population was statistically similar, but in the RS (day 154), it induced a higher population (706 × 10^4^ CFUs g^−1^) compared to the VS. The presence of CRO in the rhizosphere of both species promoted a higher RhR population (454 × 10^4^ CFUs) compared to the rhizosphere without CRO, which had 130 × 10^4^ CFUs; that is, it was 249% higher (Table 1). On the other hand, the effect of oil promoted a higher population density of rhizobia (454 × 10^4^ CFUs) than in soil without oil (130 × 10^4^ CFUs). The RhN population was 916 × 10^4^ CFUs in *C. pallida* established in control soil. The lowest population was found in both plant species that grew in both VSs in soils contaminated with 3, 15, 30, and 45 g/kg of CRO. The same response was observed in the RS of *C. pallida* (Table 1). The highest NN (657) formed in *C. incana* in the RS soil exposed to 45 g/kg CRO, while the lowest number (11) was found in *C. pallida* in the VS in soil at a similar CRO dose. *C. incana* formed a higher NN (320) compared to *C. pallida* (173) under the evaluated conditions. According to the phenological stage, the greatest NN occurred in the RS (381 nodules) than in the VS (113 nodules); the difference was on the order of 237% (Table 1).

### 2.2. Effects of CRO on AMFs According to the Crotalaria Growth Stage

The colonization of intraradical structures in AMFs showed statistically significant differences between the means of the evaluated treatments (Table 2). The greatest quantities of hyphae grew in soils without petroleum in both stages evaluated, in both *C. incana* and *C. pallida*. The results also highlight that statistically high values occurred in the VS in *C. incana* plants exposed to 3, 15, and 30 g/kg of soil. At the species level, *C. incana* formed a greater quantity of hyphae in the plant in soil without CRO (Table 2). The distribution of arbuscules in AMFs was more numerous (63% arbuscules) in *C. incana* established in control soil in the VS. In contrast, only six arbuscules were found in *C. pallida*, in the RS, exposed to 30 and 45 g/kg of soil. Overall, the legume *C. incana* formed 31% arbuscules and *C. pallida* only 17%; according to the stage, the greatest number was found in the VS with 39% arbuscules in the control soil compared to 80% in oil-contaminated soil. Data regarding vesicles within the root showed statistical similarity to those corresponding to arbuscules. In the control soil, 70 and 77% vesicles formed in the VS and RS, respectively, for *C. incana*. At the species level, 51% vesicles were counted in *C. incana* and 34 in *C. pallida*; according to the phenological stage, the number was greater in the VS with 29% vesicles, and according to the oil contamination, 63 arbuscules were found in *C. incana* and 37% were in the contaminated soil: a difference of 70% (Table 2). Spore formation in AMFs was more sensitive than in hyphae, vesicles, and arbuscules.

The values of 78 to 82% intraradical spores in the vegetative stage of *C. incana* exposed to 0, 3, and 1 g/kg of soil are noteworthy. The greatest inhibition was observed in *C. pallida* in the RS with 14% spores, which was equivalent to 17% compared to 82% spores in *C. incana* exposed to 3 g/kg of soil in the VS. In general, the results with the highest spore counts corresponded to the species *C. incana*; regarding the phenological stage, it was the VS and in soil without oil (Table 2). Regarding the presence of spores in the rhizosphere, the data were variable; the highest quantity was 1905 in *C. incana* in soil without oil, and the lowest value was 753 in *C. incana* in the RS in soil with 45 g oil/kg of soil. The general statistical analysis showed that the largest quantities of rhizospheric spores were formed in *C. incana* (1264 spores); both phenological stages accumulated quantities with statistical similarity (1180 and 1206); finally, in soil with oil, a general average of 1450 spores was found versus 1129 in contaminated soil; both data represent a difference of 28.4% (Table 2).

### 2.3. Effects of Petroleum Toxicity on Rhizobium, Nodule and Arbuscular Mycorrhizal Fungy

Hormesis in rhizosphere Rhizobium population (RhR) for both Crotalaria species showed marked differences due to plant exposure to crude oil (Table 1). The data revealed high stimulation at all four oil doses, which was particularly higher (11.1 and 11.5) during the VS in both species (Table 1). The hormetic effect (IT > 1) was evident in 13 of the 16 treatments evaluated. The overall values indicate that the highest hormesis values corresponded to both species (6 and 5.6), during the VS (8.3), and in soil treated with oil (3.5); all values were >1. Hormesis in Rhizobium nodules was less pronounced compared to rhizosphere data. Experimental data show high values in the VS of *C. pallida* exposed to 15, 30, and 45 g CRO/kg of soil, with IT values of 2, 7.3, and 8.7, respectively. As for the inhibition effect, the CRO was observed on the NNs (Table 1). The toxicity index (IT < 1), showing the inhibition by CRO dose in percentage of hyphae, was obtained in 16 treatments, and hormesis was observed only with values of 1.1 in the VS for both species planted in soil with 3 g/kg. Intraradical structures such as arbuscules, vesicles, and spores in soil were inhibited in their development with the presence of CRO, reaching IT < 1 in all 16 treatments evaluated. The stimulation or hormesis (IT > 1) was shown only in the percentage of intraradical spores in both species in the VS in soil with low doses (3 g/kg PC) (Figure 1a), but doses of 3, 15, 30 and 45 g/kg induce stimulation (IT > 1) in *C. incana* RS with values between 1 and 1.3 (Figure 1b).

### 2.4. Removal of Total Hydrocarbons from Crude Oil

The removal of the HTPs results in the treatments with *C. incana* and *C. pallida* at the VS (day 60) and RS (day 154) showing statistically significant differences (Duncan’s <0.05) (Figure 2a). The highest TPHs removal (77%) occurred in soil with 45 g/kg treated with *C. incana* rhizosphere on day 154, while the lowest removal (35%) occurred in soil with 3 g/kg in the *C. incana* rhizosphere on day 60 (Figure 2a). Depending on the species, removal was highest in the rhizosphere space of *C. incana* (64%) compared to 59% for *C. pallida*; these two species showed statistically significant differences (Figure 2b). The comparison of HTPs removal between the two phenological stages showed that in the RS, 64% of oil was removed (Figure 2c); with respect to the amount of oil added to the soil, the removal fluctuated from 60 to 61.5% in treatments with 15 and 45 g/kg of CRO, respectively (Figure 2d).

### 2.5. Removal of Linear Alkane from Crude Oil

The individual removal of each of the 15 alkanes (C12 to C26) showed statistically significant differences between treatments due to the effect of the CRO dose, Crotalaria species, and phenological stage (Duncan *p* ≤ 0.05) (Table 3). The highest mean removal (98 to 100%) occurred in the alkanes with the lowest molecular weight (C12 to C14) in treatments with 3, 15, 30, and 45 g/kg of CRO in both species during the reproductive stage. In general, it was found that the removal of alkanes gradually decreased starting with the C15 alkane; this trend did not occur in *C. pallida* during the flowering and fruiting RS. The overall analysis shows that the removal of the 15 LAs was greater in *C. pallida*. The same situation occurred during the flowering–fruiting phenological stage, and regarding the amount of CRO, the most efficient removal was in plants established with 15 and 45 g/kg of CRO (Table 3).

The greatest removal of TLAs (97 and 98%) occurred in the rhizosphere of *C. pallida* in the RS contaminated with 3 and 15 g/kg of CRO, respectively, whereas the TLAs were lowest (58%) in the rhizosphere of *C. incana* contaminated with 3 g/kg oil (Figure 3a). In general, 91% removal of TLA was observed for *C. pallida*, which was 4.4% greater than that for *C. incana* (Figure 3b). Compared with the VS (84%), greater removal (94%) occurred in the RS: an increase of 10.6% (Figure 3c). The highest removal percentages of 93, 92, and 92% were observed in the rhizospheres with 15, 30, and 45 g/kg CRO, respectively, compared with the rhizosphere with 3 g/kg oil, with 80% removal during the VS of the two Crotalaria species (Figure 3d).

### 2.6. Linear Alkanes Removal and Microbiological Association

Figure 4 contains the canonical correspondence analysis (ACC) that shows a total explained variation of 93.48% (83.42 + 10.06), where the microbial symbiotic activity (hyphae, vesicles, arbuscules, spores-rhizosphere and total colonization AMFs) was directly related to the removal of linear alkanes of lower molecular weights (C12 to C18); RhN is associated with the removal of LA C24 and C26, while RhR induced the removal of alkanes C19, C21, C22, C23 and C25. However, the removal of TLA showed a significant and positive correlation with RhR = 0.354 **, and it was negatively correlated with the AMFs structures: hyphae = −0.316 *, arbuscles = −0.606**, intraradical spores = −0.484 **, and total colonization = −0.464.

## 3. Discussion

Rhizobia and AMFs are versatile microbes that become established in the rhizosphere and roots of legumes and are adapted to degraded soils; they have been used as indicators of toxicity and for the phytoremediation of soils contaminated with CRO [27,28]. The results of our study indicate that the rhizosphere of both legumes exhibits variable effects on the root–*Rhizobium* relationship, NN, and AMFs association in contaminated soil. This is explained by the presence of different concentrations of CRO, which includes long-chain hydrocarbons that are recalcitrant and hydrophobic. In soil, it can be present as individual molecules in gaseous, liquid, or solid states, dissolved in water, adsorbed on surfaces, or absorbed into the organic soil matrix [29], and it can cause toxicity and hormesis in free-living and mutualistic rhizospheric microorganisms [8,9].

The rhizospheric system of *C. incana* is more versatile than that of *C*. *pallida*, as it is colonized by a larger population of RhR under stress conditions due to the presence of CRO, with IT = 6.0 compared with 5.6 in *C*. *pallida*. This increase in the RhR indicates dose stimulation (hormesis), which is possibly because the root of *C*. *incana* has a greater capacity to loosen the compaction induced by CRO in contaminated soil, and the root forms pores and fissures for the entry of oxygen and integrates an oxygenated rhizosphere and nutrient substrates compatible with Rhizobia [30]. The root systems of the two *Crotalaria* species in the RS had a 90% greater population of RhR than those in the VS, which was perhaps because mature roots exude greater amounts of plant hormones (jasmonate, ethylene, abscisic acid (ABA), and/or salicylate), including stress hormones that function in the regulatory mechanisms and the generation of signals for plant growth and production under anthropogenic conditions [31].

According to [32], nodular *Rhizobium* strains and non-nodular Rhizobial species exist in the soil and rhizosphere. In this context [33], only 50% of rhizobia in the soil can form nodules in legumes, and many autochthonous rhizobia are not completely effective at N_2_ fixation despite having the ability to nodulate. In our research, the 71% increase in free-living rhizobia in the rhizosphere due to the effect of CRO and the significant positive correlation between RhR and NN = 0.560 ** were noteworthy, indicating the high viability of rhizobia in the rhizosphere by the establishment of genetic signals that induce the formation of nodules in *C*. *incana* and *C*. *pallida*. We compare our results obtained for the legumes *Crotalaria incana* and *C*. *pallida* with the results of authors [27,34] who explored the effects of petroleum and its derivatives on the legumes *Arachis hypogaea* and *Phaseolus vulgaris*, *Medicago sativa*, and *Trifolium repens*, respectively. They found that used automotive engine oil has negative effects on the amount of *Rhizobium* in the rhizosphere of *A*. *hypogaea* and *P*. *vulgaris*. The second author reported a drastic decrease in endophytes due to plant exposure to total petroleum hydrocarbons. The increase in the abundance of RhR of the *Crotalaria* species, evaluated in this study, is possibly due to the genetic development of these endophytes to use various sources of carbon, nitrogen, and energy of biogenic origin (sugars, sugar alcohols, carbohydrates, amino acids, urea, nitrate, and N_2_) [35] and petroleum (linear, branched, and aromatic polycyclic saturated hydrocarbons) [36] for cellular metabolism.

*Rhizobium*–legumes symbioses are controlled by the chemistry of the flavonoids secreted by each species, although they are planted in similar soil. The attraction of rhizobia (quimiotaxis) produces lipo-chitooligosaccharides (*Nod* factors), which will penetrate the root hairs via infection strands to form swellings or nodules and subsequent bacteroids growth (N_2_-fixing nodule bacteria) [37]. In our research, NN was positively correlated with the population of RhN = 0.670 **, where *C*. *incana* reached a 46% greater nodulation than *C*. *pallida*. Another important aspect in the decrease in nodule formation in some legumes is due to the low content of water and oxygen, the loss of nutrients, the decrease in microbial activity, and the increase in heavy metal contents as a consequence of CRO in the soil [38].

In our research, the greater nodular capacity of *C*. *incana* is possibly due to the intense horizontal signal transfer of genes such as *nod* and *nif*, which are located on plasmids or in transferable regions of the chromosomal DNA of rhizobia [16], and they are responsible for the stimulation of nodulation (*Nod* genes) [39] despite the presence of stress conditions. The nodules of some legumes can keep *Rhizobium* viable, but there are also nonviable nodules because of the lack of leghemoglobin responsible for transporting oxygen in the root nodules that work in symbiosis with *Rhizobium* [5].

Compared with *C*. *pallida*, *C*. *incana* contains 63% (Table 1) more CFUs g^−1^ of *Rhizobium*, which indicates that the infected root cells of *C*. *incana* synthesize leghemoglobin to bind oxygen, and they also facilitate diffusion to bacteroids while maintaining the concentrations of oxygen necessary for respiration and the fixation of N_2_ to ammonia (NH_3_), making it available for the plant [40]. The RS of the two *Crotalaria* species induced a 70% increase in the NN and a 97% increase in the CFUs g^−1^ RhR with respect to the VS. According to [33,41], in perennial plants, aerial vegetative organs stop growing, but the root can always continue growing so that the demand for N increases due to root growth, and to satisfy the need for N supply, the root maintains the exit signal from the initiation and nodule formation pathway.

The binomial NN and the RhN increased in the RS of *C*. *incana* and *C*. *pallida*, indicating that the plants not only provided electrons for the synthesis of gibberellin for flowering and carbohydrates for fruit formation [42] but also required electrons to be provided to the RhN for N fixation [15]. In the RS, there is clearly a cross-transfer of legumes–*Rhizobium* genes with greater compatibility, and phenolic compounds [flavonoids or isoflavones (luteolin, genistein, daidzein, and naringenin)] may be released from the root at this stage of growth along with nonflavonoids (betaines, xanthones, aldonic acids, jasmonates, and phenolic compounds) and proteins (lectins, trifolins, and remorins) [36] that bind to a rhizobial gene product encoding the *Nod* transcription factor to induce the curling of root hairs and trigger root cell division, which ultimately leads to the formulation of nodules in greater quantity associated with the mature phase of growth.

The decrease in NN is similar to that mentioned by [27], who observed NN levels that were 77, 86, and 49% lower in *A. hypogaea*, *P. vulgaris* (White), and *P*. *vulgaris* (Brown), respectively, in soil with 3% motor oil than in the control; also, other studies indicated a 26% lower NN associated with *P*. *vulgaris* in soil with 3 g/kg TPHs, and NN decreased by 30% in *Clitoria ternatea* exposed to 6 g/kg CRO [43,44].

The reduced CFUs g^−1^ counts of viable RhN in the presence of localized oil in our investigation was similar to [45] in nodules of *A. hypogaea* exposed to 1% diesel, but a total inhibition of the bacteria was observed in soil with 4 and 8% diesel. The inhibition of NN and RhN in the nodule is caused by (a) the hydrophobic conditions induced by the CRO in the soil that restrict the water and oxygen content for the plant, which affects the bacterial multiplication within the nodule, and (b) the decrease in the synthesis of nitrogenase due to hydric stress such that *Rhizobium* and NN decrease because the nodules detach and die when the plant allows electrons to be sent to the nodule [46,47]. Similarly, the authors of [39] found that the ability of some legumes to counteract the stress induced by petroleum hydrocarbons limits the ability of *Nod* factor receptors to recognize and respond to *Rhizobium*-specific *Nod* factors and stimulate the ripple of root hairs and cortical cell division to form nodules [5,48]. These authors explained that plants under stress conditions prioritize the synthesis of antioxidants involved in counteracting reactive oxygen species, which allows them to maintain cells and limit death, thus possibly limiting the genetic signals of symbiotic associations with *Rhizobium.*

Previous studies have shown that the association of native AMFs with plants is affected by environmental factors such as the species of plants, their life cycle (annual or perennial), their fertility, and the type of soil [49], as well as stress factors such as temperature, pH, soil moisture, root exudates, and the presence of organic pollutants, such as petroleum hydrocarbons [33,50]. In this study, it was observed that *C*. *incana*, with respect to *C. pallida*, induces a greater percentage of colonization of four intraradical structures and spores in AMFs soil; similar results [51] in *V. sativa* and *V*. *narbonensis* were established under similar soil conditions: *V*. *sativa* showed a greater colonization of AMFs species. In this regard, it is stated that a similar pattern of seasonal mycorrhizal colonization index should not be expected in plants, especially when their root systems, growth periods, and dependence on mycorrhizae to grow in nutrient-poor soils differ [52].

In our research, the growth period influenced AMFs colonization with the VS (day 60) having a greater percentage of hyphae, arbuscules, intraradical spores, and spores in the rhizosphere compared with the RS (day 154). One explanation may be that in the initial stage of plant establishment in the soil, they seem to have a greater affinity and dependence on AMFs, which is mainly because of their high growth rate and root production and the differentiation of vegetative organs [53]. Under conditions of abiotic stress, mycorrhizal structures can change the dynamics of colonization in plant roots; in this study, the negative effect of CRO on AMFs was observed at concentrations of up to 45 g/kg oil with IT < 1 for hyphae, vesicles, arbuscules, and spores in the rhizosphere at both the VS and the RS of the two *Crotalaria* species, and the intraradical spores were notably stimulated mainly in the RS of *C*. *incana* under the four CRO doses evaluated in this study. Hyphae decreased in association with *Cichorium intybus* at concentrations of 35, 70, 140, and 280 µM benzo(a) pyrene [54]. In this regard, it is confirmed that the native AMFs communities exhibit decreased colonization in the roots of plants growing in contaminated soils compared with those in soils without contamination [55,56].

Oil doses between 3 and 45 g/kg induce a decrease in AMF colonization, which can be explained by the fact that CRO prevents the passage of water, nutrients, and oxygen in the soil aggregates and the roots, affecting AMFs establishment and multiplication. This decreased colonization can also be attributed to the fact that the cortical cells of *C*. *incana* and *C*. *pallida* subjected to stress contain small amounts of sugar, which AMFs depend on to carry out their metabolic processes [57]. The low amount of photosynthates represents a disadvantage in terms of the adaptability of the plant. In addition, AMFs use between 4 and 20% of the carbon fixed (sugars) by the plant during photosynthesis [14].

The inhibition of the NN, RhN, hyphaes, arbuscles, vesicles, and spores–soil of AMFs in the VS and RS of *C. incana* and *C. pallida* was observed at doses of 3, 15, 30, and 45 g/kg CRO using the dose–response index toxicity method with the ratio of contaminated treatment to control. The results are reliable because a IT < 1 indicates a negative effect of CRO on the microbiological variable, and although it is an absolute value, it changes according to the CRO dose. This behavior is similar to the sensitivity trend of *Vibrio fischer* to increasing doses of 3,5-dichlophenol (EC20, EC50, and EC80) as determined by toxicity tests modified by [58] and standardized by the ISO and OECD. The inhibition of intraradical AMFs structures with similar methods used in this investigation was reported by [8] in *Leerxia hexandra* roots exposed to different doses of CRO [9]. An IT > 1 was reported in *Pseudomonas* spp. exposed to between 3 and 75 g/kg of CRO in the rhizosphere of *Eleocharis palustris*.

Highlights include the following: the increase in intraradical spores with IT > 1 in *C*. *incana* in the RS due to the effect of CRO is similar to that claimed by [8] in the grass *L. hexandra* in soil with 30 g/kg oil [59]. The colonization of AMFs in the roots of *Persicaria lapathifolia*, *Lythrum salicaria*, *Lycopus europaeus*, and *Panicum capillare* in sediments contaminated with oil is greater than that in uncontaminated sediments. This positive response can be explained by the possibility that *Crotalaria* in the RS is capable of assigning to the roots a substantial proportion of the photosynthates necessary for the maintenance of AMFs multiplication due to the age of the plant [15]. The highest percentage removal value according to the oil dose was 68% in soil with 45 g; however, the *C. incana* treatment with 45 g/kg CRO on day 154 (RS) exhibited 76.6% removal. These results are similar to those mentioned by [60], who found that on day 90, a 59% removal of TPHs derived from soil with 6120 mg/kg weathered CRO was observed for *Lotus corniculatus* L. The authors of [61] found a 59% removal of TPHs induced by the *Leucaena leucocephala* rhizosphere in soil with 50 g/kg oil on day 158. The authors of [62] mentioned that after four months of exposure to oil sludge in soil, the TPHs were reduced by 57.4% for *Cannavalia ensiformis*. The authors of [63] reported a 56% removal of TPHs induced by *Vigna radiata* planted in soil with 5 g/kg of diesel.

The greater growth of the root system in RS in *C. incana* than in *C. pallida* can increase the population of microorganisms because of compounds that promote oil degradation. However, in our research, a similar trend occurred: a highly significant positive correlation was observed between TPHs * RhR = 0.443 **, while there was a negative correlation with RhN = −0.622 ** and the percentage of arbuscules = −0.374 **. This positive relationship has been previously mentioned, where the degradation of hydrocarbons in the rhizosphere and soil by free-living bacteria is attributed to their ability to synthesize surfactants that break the surface tension between water and oil, making hydrocarbons available for degradation [15].

Since then, it has been mentioned, for example, that *Rhizobium* in soil contaminated with 2% and 4% diesel removed the pollutants, with total removal percentages of 54 and 50% for *V. unguiculata*, 68.8 and 52% for *V. faba*, 52 and 59% for *Phaseolus*, and 65 and 47% for *V. radiata*, respectively [64]. The 15 linear alkanes were removed in greater quantities by the rhizosphere of *C*. *pallida*: between 99% and 88% for C12 to C20 LAs and between 85 to 64% for C21 to C26 Las. The rhizosphere of *C*. *pallida* induced 91% greater LAs removal (ƩC12–C26) with respect to *C*. *incana* at the end of the experiment. Similar results were mentioned by [65], who found that *Medicago sativa* was used in soil with 30% oily sludge, the LAs removal efficiency for C14–C38 was 78.46%, for C13–C16 was 89.17%, and for C17–C18 was 78.26%, respectively. It is claimed that the removal efficiencies of 78.6 and 48.9% for the rhizospheres of *M. sativa* and *Festuca arundinacea*, respectively [66]. Our results reveal that short-chain alkanes were removed at a greater percentage than long-chain alkanes; these results are similar to those of [67], who found 45.8% removal (C8–C40) in soil contaminated with 7719 mg/kg CRO planted with *Lolium* spp., but they are in contrast to the findings of [68], who mentioned 100% removal (C30–C33) of oil in the presence of *M*. *sativa* after 60 days of exposure.

In this regard, according to [69], light hydrocarbons can be easily volatilized and oxidized, whereas heavier hydrocarbons can adhere to the organic matter of the soil, leading to greater toxicity and less degradation depending on the concentrations present in the soil and the exposure time. In our research, the CRO exposure time of 60 to 154 days influenced the rhizosphere of the *Crotalaria* species such that the removal of C12 to C21 LAs increased between 2 and 16% and between 21 and 38% for LAs of greater molecular weight (C22 to C26). Likewise, 94% of the TLA was removed by day 154 (RS), and the removal of TLA was 14 and 13% higher in soil with 15, 30, and 45 g/kg of oil, respectively, with respect to that in soil with 3 g/kg of oil. In this regard, indicated that a reduction in the concentration of n-alkanes with C12–C18 can occur after 60 days of applying biological technologies for cleaning soil contaminated by CRO [70], and its is mentioned that the degradation of long-chain n-alkanes (C25–C30) via oxidation reactions [71]. According to [15], under toxic conditions with the availability of essential nutrients for microorganisms, 80% or more of nonvolatile petroleum compounds can be oxidized over the course of one year. However, the high removal of LAs is related to their chemical structure, considering that short-chain LAs (C10–C24) are easily degraded via the metabolic processes of bacteria, but the hydrophobicity increases with chain length, and long-chain LAs tend to be more attached to organic matter, which may hinder their degradation [72]. 

In our investigation, a positive RhR relationship was found with the removal of alkanes of higher molecular weight (Figure 4), which is similar to the results mentioned by [73] with *Pseudomonas aeruginosa*, which, when inoculated in soil contaminated with diesel (10% *v*/*v*), completely degraded C16, C17, C18, and C22 LAs. Other studies evidenced that *Acinetobacter calcoaceticus* was able to remove between 82 and 92% of C12 and C18 LAs after 28 days of exposure to diesel [74], whereas *P*. *aeruginosa* removed 80 and 98% of C16 and C19 LAs, respectively, after seven days of exposure to CRO [75].

The increase in the RhR and the decrease in AMFs colonization in the rhizospheres of *C. incana* and *C. pallida* contaminated with CRO are noteworthy because the first group is positively correlated with TPHs and LAs, which have higher molecular weights; however, the second group (the percentage of colonization with vesicle, hyphae, arbuscules, and spores) has a positive relationship with LAs of lower molecular weight, but they are limited in degrading molecules of higher molecular weight. Therefore, we assume that in the environment of stress caused by oil, microorganisms establish a tripartite legumes–*Rhizobium*–AMF synergy [76]. In this tripartite relationship, the extraradical mycelia of AMFs can provide the necessary nutrients for the colonization and growth of rhizobia and plant populations [77]. Therefore, in this research, AMFs synthesizes alkane monooxygenase enzymes that hydroxylate lower-molecular-weight LAs as a primary source of carbon, producing primary alcohols, although fungi can metabolize hydrocarbons with another carbon source [15], while the Rhizobia population increases in the rhizosphere from VS to RS (33.8 ×10^4^ CFUs g^−1^) using organic carbon derived from the mycelia of AMFs as a primary source of energy [78] and subsequently synthesizes monooxygenases to degrade high-molecular-weight LAs, which is a phenomenon called cometabolism [15].

## 4. Materials and Methods

### 4.1. Soil, Plants, and Crude Oil

The soil used for the study was an uncontaminated clay loam soil collected from the surface horizon (0–30 cm) of the Paso y Playa common land, which is located at km 2 of the Federal Highway Cárdenas-Huimanguillo (17°58′08″ N and −93°36′ 09″ W), Cardenas, Tabasco, Mexico. The soil was dried indoors and sieved through a mesh (5 mm aperture). The properties of the soil were as follows: pH 5.5, 0.5 dS/m electrical conductivity, 6.1% organic matter, 3.6% organic carbon, 0.13% Nt, and 1.024 g/kg TPHs on a dry basis.

Two legumes of the *Crotalaria* species (*incana* and *pallida*), perennial plants of the humid tropical region, were used (Figure 5b,f). The roots of these legumes are pivotally branched, resulting in the establishment of double mutualism, with *Rhizobium* forming root nodules and fungi developing AMFs. For the purposes of this research, the rhizosphere is defined as the root colonized by symbiotic microorganisms and the adhered soil (Figure 5a). The two *Crotalaria* species have trifoliate compound leaves (Figure 5c,g), yellow flowers (Figure 5d,h), and pod-shaped fruits, which are pubescent in *C*. *incana* (Figure 5e) and smooth in *C*. *pallida* (Figure 5i). The seeds of *C*. *incana* and *C*. *pallida* were collected in La Venta Tabasco, México. The CRO used was medium crude, with a 25.9° API and a specific gravity of 0.84 g/cm, 56.4% aliphatic fraction, 23.7% of aromatic fraction, and 14% asphaltenes and resins. The oil was obtained from the Cinco Presidents oil field located in La Venta, Tabasco, Mexico (18°12′11.8″ N and −94°08′37.8″ W).

### 4.2. Experimental Design

A factorial experiment with a completely randomized 2 × 2 × 5 design and four replicates was performed in a microtunnel. Two *Crotalaria* species, two growth stages (VS, day 60) when more than 50% of the stems presented bifurcation, and the reproductive stage (RS, day 154, when more than 50% of the branches presented flowering and fruiting), and five doses of CRO [0 (control), 3, 15, 30, and 45 g/kg of soil]. The experimental unit consisted of a glass pot (14 cm high × 18 cm in diameter) with 2000 g of soil dw. The CRO was weighed for the appropriate dose (g/kg) and incorporated into the soil. On day 2 after the soil + CRO mixture was prepared, a soil sample was taken for the initial evaluation of TPHs and 15 LAs (C12 to C26), irrigation was applied, and a *Crotalaria* seedling was sown.

### 4.3. Nodules and Population of Microorganisms

The NN was evaluated by direct counting. The population of endophytic RhR and in the RhN were identified as CFUs on the basis of their morphology according to Bergey’s Manual of Systematic Bacteriology [32]. The technique was performed by serial dilution in 1 g of nodule for RhN and 10 g of rhizosphere for RhR in specific culture medium, yeast extract–mannitol–agar–Congo red. The plates were incubated at 30 °C for 48 h. The colonization of AMFs (intraradical hyphae, arbuscules, vesicles, and spores) in the root was studied using the clearing and staining method of [79], and the percentage of roots colonization was calculated for each fungal structure using the formula MC% = (NCR/NRS) (100), where MC% is the degree of AMFs colonization, NCR is the number of colonized roots, and NRS is the number of root segments observed. The spore count in the rhizosphere (25 g) was carried out under the sieving and decantation methodology of [80] with centrifugation in a sucrose gradient (20 and 60%) [81].

### 4.4. Index of Toxicity

The index of toxicity (IT) integrates several variables and provides a value that can be used to holistically determine the NN and microbiological response in soil contaminated with different concentrations of CRO. The IT was calculated for variable (X).

The IT of each variable was compared with the corresponding value of the control treatment (0 g/kg CRO). The formula used was as follows:

where

IT: Index of toxicity;

Xi: Variable i;

CT: Control treatment;

CROT: Crude oil treatment;

ITxi = CROT/CT………………i 1,2, …, n.

The results were interpreted as follows:

>1 indicates that the variable was stimulated by CRO (hormesis);

1 the variable was not inhibited by CRO (tolerance);

<1 the variable was inhibited by CRO

### 4.5. Analysis of Total Petroleum Hydrocarbons and Linear Alkanes

The TPHs were extracted for 8 h in Soxhlet equipment [82] with analytical grade dichloromethane. The coefficient of variation was less than 9.2%. The amount of TPHs was calculated gravimetrically after concentrating the oil with a rotary evaporator.

The concentrate obtained for TPHs was used for the determination of 15 LAs (C12 to C26). The sample obtained was dissolved in 1.0 mL of dichloromethane for filtration to eliminate particulate fines that would interfere with the chromatographic analysis. N-25-4 nylon isodisc filters (25 mm × 0.45 μm in diameter) were used. The clean filtrate was brought to a volume of 1.0 mL and injected intro gas chromatography coupled with mass spectrometry (GC-MS) [83]. To determine the 15 LAs (C12 to C26), 2 μL of the extract was injected into an Agilent Technologies™ gas chromatograph (Model 6890 N; Net Work GC system) equipped with a DB-5 column (5% phenylmethyl polysiloxane), which was 60 m long, 0.25 mm in internal diameter, and 0.25 μm in film thickness. The initial temperature was 50 °C and was maintained for 5 min; the temperature was then increased at a rate of 20 °C/min to 280 °C, which was maintained for 20 min. Helium was used as the carrier gas with a flow rate of 1 mL/min, and the injector temperature was 250 °C.

Once the chromatograms were obtained, LAs identification was carried out via mass spectrometry using an Agilent Technologies Model 5975 inert XL mass spectrometer (Santa Clara, CA, USA). Mass spectra were obtained by electron impact ionization at 70 eV, and the mass spectra obtained for each compound were compared with a database (HP Chemstation-NIST 05 Mass Spectral search program, version 2.0d) and with a standard (saturated alkanes, C7-C30; catalogue no. 49451-U; Sigma‒Aldrich, Darmstadt, Germany), which was determined under the same conditions. The concentration (ppm) of each of the LAs was emitted in each chromatogram by the HP mass spectrometer.

### 4.6. Removal of Total Petroleum Hydrocarbons and Linear Alkanes

The effect of phytoremediation of TPHs, 15 LAs (C12 to C26) and total linear alkanes (LAT) at days 60 and 154 was determined by the difference in concentration at day 1 compared to the values from days 60 to 154 after the experiment was established.

### 4.7. Statistical Analysis

Full factorial experiments in CRD with specified treatments and repetitions were carried out for all investigations. For data collection and analysis, three-way analysis of variance (three-way ANOVA, R Core Team, Vienna, Austria) was applied carefully. The statistical clearance was determined at a 95% confidence level and through using Duncan’s post hoc test (*p* ≤ 0.05). Canonical correspondence analysis (CCA) and Pearson correlation were also performed between the microbial symbiotic activity and the removal of total and linear hydrocarbons from the CRO.

## 5. Conclusions

CRO was toxic at doses between 3 and 45 g/kg in the rhizospheres of *C. incana* and *C. pallida* because it decreased the population of *Rhizobium* in the nodules, the number of nodules, and the intraradical AMFs structures but had a positive effect on the population of Rhizobia in the vegetative and reproductive stages of the two Crotalaria species. The biological parameters of *Rhizobium* in the nodules, the number of nodules, and the percentage of hyphae, arbuscules, vesicles, and spores in the rhizosphere are ideal for use in CRO toxicity tests, and Rhizobia is ideal for uses in the rhizosphere hormesis test. The removal of TPHs, individual linear alkanes of higher molecular weight, and total linear alkanes was greater in the RS than in the VS, and these removals were significantly positively correlated with RhR. A tripartite rhizosphere relationship of legumes–*Rhizobium*–AMFs was observed, where RhR increases while AMFs colonization decreases possibly to sustain Rhizobia growth and increase hydrocarbon removal. The rhizosphere of *C. incana* on day 156 (RS) has the potential to remediate soil contaminated with up to 45 g/kg CRO by removing 77% of the TPHs; however, the rhizosphere of *C. pallida* stimulates the removal of between 99 and 88% of C12 to C20 LAs, between 85 and 64% of C21 to C26 LAs, and 91% of the total linear alkanes. The number of nodules, the *Rhizobium* in the nodules, and the percentage of AMFs on days 60 (RV) and 156 (RS) should be used in toxicity tests, and phytoremediation technologies should be used to determine the TPHs and LAs using the rhizosphere of *C. incana* and *C. pallida*, respectively, in loamy-clay soils contaminated with oil in the humid Mexican tropics.

## Figures and Tables

**Figure 1 plants-15-00103-f001:**
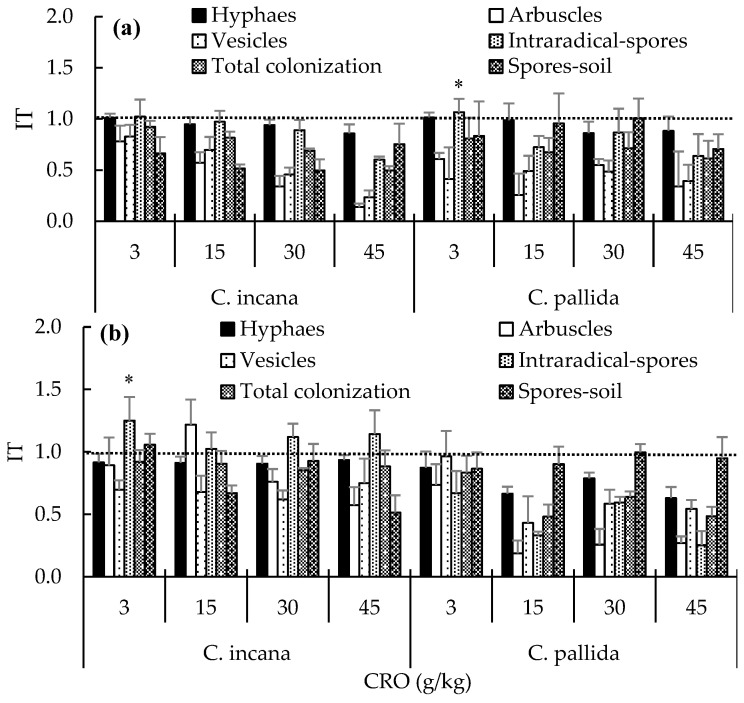
Crude oil dose–response index of AMFs structures associated with *C. incana* and *C*. *pallida*. (**a**) Vegetative stage and (**b**) reproductive stage. IT: index of toxicity. Means (±standard error): * Statistically higher index of six variables in each phase according to Duncan’s test (*p* ≤ 0.05). Value > 1 indicates that the variable was stimulated by CRO (hormesis). Value = 1 indicates that the variable was not inhibited by CRO (tolerance). Value < 1 indicates that the variable was inhibited by CRO.

**Figure 2 plants-15-00103-f002:**
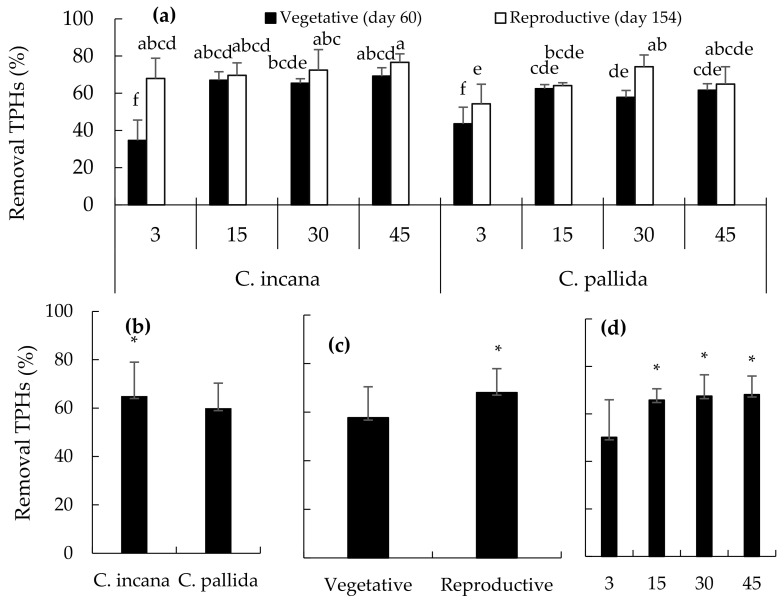
Total petroleum hydrocarbon removal (**a**) by treatments (**b**) according to the type of *Crotalaria*; (**c**) according to growth stage; (**d**) according to the crude oil dose. Means (±standard error) followed by the same letter in the column do not differ significantly according to Duncan’s test (*p* ≤ 0.05). * Means by statistically significant factor.

**Figure 3 plants-15-00103-f003:**
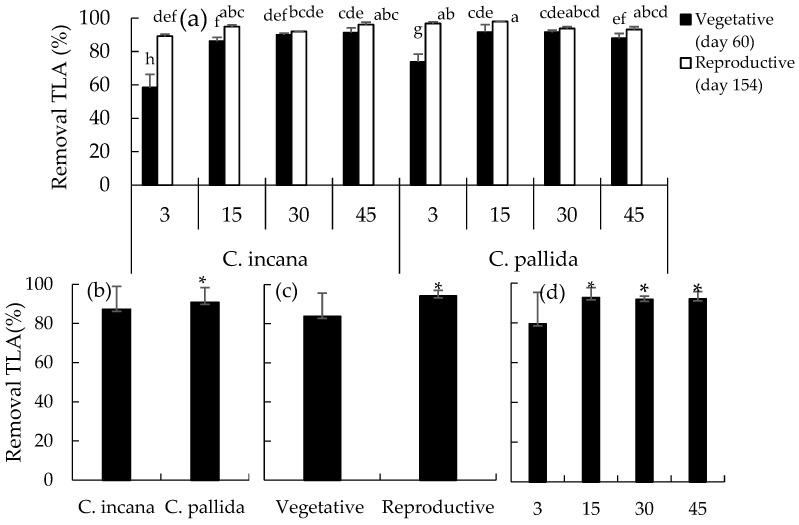
Total linear alkanes (TLA) removal (**a**) by treatment, (**b**) according to the species of *Crotalaria*, (**c**) according to vegetative stage, (**d**) according to the crude oil dose. Means (±standard error), when followed by the same letter in the in the column, do not differ significantly according to Duncan’s test (*p* ≤ 0.05). * Means by statistically significant factor.

**Figure 4 plants-15-00103-f004:**
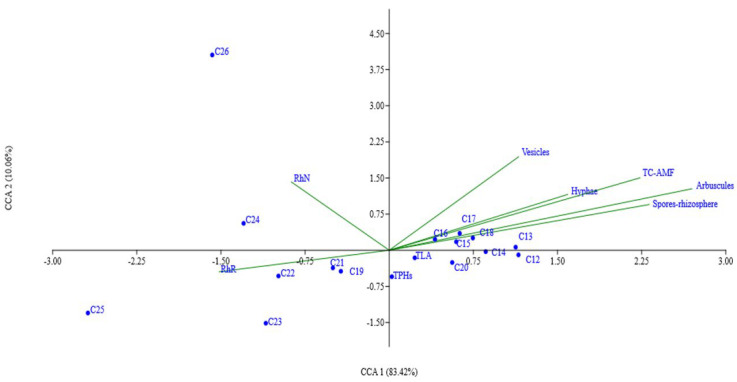
Canonical correspondence analysis between hydrocarbon removal and microbial symbiotic activity in rhizospheres of *C*. *incana* and *C*. *pallida* exposed to crude oil. RhN: rhizobium in nodule. RhR: rhizobia in rhizosphere. TLA: total linear alkanes. TPHs: total petroleum hydrocarbons. TC-AMF: total colonization arbuscular mycorrizal fungi.

**Figure 5 plants-15-00103-f005:**
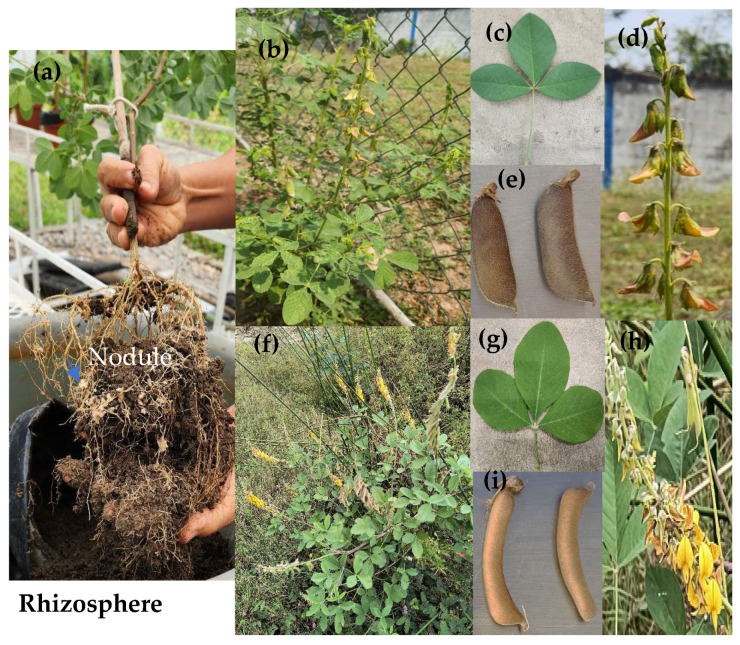
Rhizosphere (**a**), morphology of the legumes *C. incana* (**b**) and *C. pallida* (**f**) growing in a microtunnel but with seed collected in La Venta, Tabasco, Mexico. Leaves, flowers and fruit of *C. incana* (**c**–**e**) and *C. pallida* (**g**–**i**).

**Table 1 plants-15-00103-t001:** Effect of treatments, factors, and index of toxicity for crude oil, including the Rhizobia in rhizosphere, *Rhizobium* in nodule and nodules number.

T	*Crotalaria*/Stage	CRO (g/kg)	(10^4^ CFUs g^−1^)		NN	IT		
			RhR	RhN		RhR	RhN	NN
1	*incana*/VS	0	13 ^i^	8 ^g^	182 ^efg^	----	----	----
2		3	18 ^i^	1 ^g^	203 ^efg^	1.4	0.1	1.1
3		15	144 ^fgh^	11 ^g^	184 ^efg^	11.1	1.4	1.0
4		30	144 ^fgh^	3 ^g^	78 ^h^	11.1	0.4	0.4
5		45	150 ^fgh^	3 ^g^	54 ^hi^	11.5	0.4	0.3
6	*pallida*/VS	0	8 ^i^	3 ^g^	166 ^g^	----	----	----
7		3	79 ^hi^	3 ^g^	162 ^g^	9.9	1.0	1.0
8		15	89 ^ghi^	6 ^g^	63 ^hi^	11.1	2.0	0.4
9		30	11 ^i^	22 ^g^	28 ^hi^	1.4	7.3	0.2
10		45	74 ^hi^	26 ^g^	11 ^i^	9.2	8.7	0.1
11	*incana*/RS	0	282 ^e^	916 ^a^	515 ^c^	----	----	----
12		3	218 ^ef^	620 ^b^	369 ^d^	0.8	0.7	0.7
13		15	1034 ^c^	439 ^d^	375 ^d^	3.7	0.5	0.7
14		30	1110 ^c^	275 ^e^	585 ^b^	3.9	0.3	1.1
15		45	1301 ^b^	277 ^e^	657 ^a^	4.6	0.3	1.3
16	*pallida*/RS	0	218 ^ef^	580 ^c^	470 ^c^	----	----	----
17		3	138 ^fgh^	219 ^f^	243 ^e^	0.6	0.4	0.5
18		15	196 ^efg^	33 ^g^	189 ^efg^	0.9	0.0	0.4
19		30	773 ^d^	26 ^g^	172 ^fg^	3.5	0.0	0.4
20		45	1792 ^a^	17 ^g^	233 ^ef^	8.2	0.0	0.5
Factor								
*Crotalaria*		*incana*	441 *	255 *	320 *	6.0	0.5	0.8
		*pallida*	338 *	93	173	5.6	2.4	0.4
Stage		VS	73	9	113	8.3	2.7	0.6
		RS	706 *	340 *	381 *	3.3	0.3	0.7
CRO		Without	130 *	377 *	333 *	----	----	----
		With	454	124	225	3.5	0.3	0.67

T: treatment, CRO: crude oil, RhR: rhizobia in rhizosphere. RhN: rhizobium in nodules, VS: vegetative stage, NN: number of nodules. IT: index toxicity. Means, when followed by the same letter in the column, do not differ significantly according to Duncan’s test (*p* ≤ 0.05). * Means by statistically significant factor. Value > 1 indicates that the variable was stimulated by CRO (hormesis). Value = 1 indicates that the variable was not inhibited by CRO (tolerance). Value < 1 indicates the variable was inhibited by CRO.

**Table 2 plants-15-00103-t002:** Percentage of hyphae, arbuscules, vesicles, intraradical spores and spores in rhizosphere of *C*. *incana* and *C*. *pallida* exposed to various concentrations of CRO during the vegetative (day 60, VS) and reproductive (day 154, RS) stages from March to July 2023.

T	*Crotalaria*/Stage	CRO (g/kg)	Colonization (%)				
			Hyphae	Arbuscules	Vesicles	Intraradical-Spores	Spores-Rhizosphere
1	*incana*/VS	0	95 *	63 *	77 *	80 *	1905 *
2		3	96 *	50	63	82 *	1266
3		15	90 *	36	53	78 *	984
4		30	90 *	22	35	71	947
5		45	82	9	18	48	1436
6	*pallida*/VS	0	76	40	48	44	1169
7		3	77	25	20	47	975
8		15	75	10	24	32	1119
9		30	65	22	23	38	1177
10		45	67	9	19	28	826
11	*incana*/RS	0	96 *	29	70 *	52	1463
12		3	88	26	49	65	1548
13		15	88	35	47	53	981
14		30	87	22	43	58	1357
15		45	90	17	53	59	753
16	*pallida/RS*	0	96 *	22	58	54	1264
17		3	84	16	56	36	1094
18		15	64	4	25	18	1141
19		30	75	6	34	32	1260
20		45	60	6	31	14	1199
* Crotalaria *							
*incana*			90 ^a^	31 ^a^	51 ^a^	64 ^a^	1264 ^a^
*pallida*			74 ^b^	17 ^b^	34 ^b^	34 ^b^	1122 ^b^
Stage							
Vegetative			81 ^a^	29 ^a^	38 ^b^	55 ^a^	1180 ^a^
Reproductive			83 ^b^	18 ^b^	47 ^a^	44 ^b^	1206 ^a^
CRO							
Without			91 ^a^	39 ^a^	63 ^a^	57 ^a^	1450 ^a^
With			80 ^b^	20 ^b^	37 ^b^	47 ^b^	1129 ^b^

T: treatment * Highly significant treatment means. Means per factor followed by the same letter in the column do not differ significantly according to Duncan’s test (*p* ≤ 0.05).

**Table 3 plants-15-00103-t003:** Effect of *Crotalaria incana*, *Crotalaria pallida*, growth phase, and crude oil dosage on the individual removal of 15 linear alkanes (C12 to C26).

*Crotalaria*/Stage	CRO g/kg	Linear Alkane (Carbon Chain Length)														
		12	13	14	15	16	17	18	19	20	21	22	23	24	25	26
		------------------------------------------------------------Removal (%)-----------------------------------------------------														
*incana*	3	100 *	100 *	85	75	68	74	79	30	69	30	17	5	21	22	29
(VS)	15	98	96	93	90	86	88	89	85 *	85	81	71	65	55	43	34
	30	98	97	95	94	92	87	89	91 *	90	83	78	72	67	53	57
	45	97	94	93	92	92	92	92	92 *	92	89 *	87 *	84	78	71	61
*pallida*	3	100 *	100 *	100 *	86	79	83	85	57	78	58	50	90 *	45	25	29
(VS)	15	99	97	97	95	93	92	92	90 *	88	87 *	83	75	74	74	70
	30	98	96	95	94	93	90	92	94 *	91	84 *	85 *	79	75	65	45
	45	95	91	89	89	89	90	89	89 *	88	86 *	83	79	73	66	51
*incana*	3	100 *	100 *	100 *	97	96	94	94	86 *	84	82	78	69	81	47	87
(RS)	15	100 *	100 *	99 *	99 *	96	95 *	94	96 *	94 *	92 *	82	78	86	82	72
	30	100 *	100 *	98 *	95	92	90	93	91 *	90	86 *	79	77	71	55	58
	45	100 *	100 *	98 *	98 *	98 *	95 *	95	94 *	95 *	93 *	93 *	91 *	90 *	89	7
*pallida*	3	100 *	100 *	100 *	100 *	100 *	99 *	99 *	95 *	97 *	92 *	92 *	100 *	90 *	70	95 *
(RS)	15	100 *	100 *	100 *	100 *	99 *	98 *	97 *	98 *	96 *	97 *	92 *	94 *	91 *	93 *	93 *
	30	100 *	100 *	98 *	96	94	91	95	94 *	92	89 *	87 *	79	83	76	54
	45	100 *	100 *	98 *	95	94	90	89	91 *	90	88 *	93 *	92 *	92 *	90 *	73
* Crotalaria *																
*incana*		99 ^a^	98 ^a^	95 ^b^	92 ^b^	90 ^b^	90 ^b^	91 ^b^	83 ^b^	87 ^a^	80 ^b^	73 ^b^	68 ^b^	69 ^b^	67 ^b^	60 ^a^
*pallida*		99 ^a^	98 ^a^	97 ^a^	94 ^a^	92 ^a^	92 ^a^	92 ^a^	88 ^a^	90 ^a^	85 ^a^	83 ^a^	86 ^a^	78 ^a^	70 ^a^	64 ^a^
Stage																
VS		98 ^b^	96 ^b^	94 ^b^	89 ^b^	86 ^b^	87 ^b^	88 ^b^	78 ^b^	85 ^b^	80 ^b^	69 ^b^	69 ^b^	61 ^b^	52 ^b^	47 ^b^
RS		100 ^a^	100 ^a^	99 ^a^	97 ^a^	96 ^a^	94 ^a^	94 ^a^	93 ^a^	92 ^a^	90 ^a^	87 ^a^	85 ^a^	85 ^a^	75 ^a^	76 ^a^
CRO (g/kg dw)																
3		100 ^a^	100 ^a^	96 ^a^	90 ^c^	86 ^b^	88 ^b^	89 ^c^	67 ^b^	82 ^b^	66 ^b^	59 ^c^	66 ^c^	59 ^c^	41 ^c^	60 ^ab^
15		99 ^b^	98 ^b^	97 ^a^	96 ^a^	94 ^a^	94 ^a^	93 ^a^	92 ^a^	91 ^a^	89 ^a^	82 ^b^	78 ^b^	76 ^b^	73 ^a^	67 ^a^
30		99 ^b^	98 ^b^	97 ^a^	95 ^ab^	93 ^a^	89 ^b^	92 ^ab^	92 ^a^	91 ^a^	86 ^a^	82 ^b^	77 ^b^	74 ^b^	62 ^b^	54 ^b^
45		98 ^c^	96 ^c^	95 ^b^	93 ^b^	93 ^a^	92 ^a^	91 ^b^	91 ^a^	91 ^a^	89 ^a^	89 ^a^	86 ^a^	83 ^a^	79 ^a^	66 ^a^

VS: vegetative stage (day 60). RS: reproductive stage (day 154). * Highly significant treatment means according to Duncan’s test (*p* ≤ 0.05). Means per factor followed by the same letter in the column do not differ significantly according to Duncan’s test (*p* ≤ 0.05).

## Data Availability

The original contributions presented in the study are included in the article; further inquiries can be directed to the corresponding author.

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
