# Peer review of "The Use of Rhizospheric Microorganisms of Crotalaria for the Determination of Toxicity and Phytoremediation to Certain Petroleum Compounds"

_plants, 2025, doi:10.3390/plants15010103_

Round 1

Reviewer 1 Report

Comments and Suggestions for Authors

Due to numerous errors in the manuscript, which are listed below, I suggest re-submitting the article after thorough editing and strict linguistic proofreading by a certified translator.

The quality of the figures needs improvement. Figures labeled a, b, and c should be spaced apart for better clarity. Furthermore, if the Y-axis is labeled 'removal TPHs %,' why include 'crude oil g/kg' on them?

Abstract - incomprehensible, not an introduction in a condensed form; The last sentence is too far-reaching (instead of 'should,' it is more appropriate to use 'might')

Results

  1. Incorrect CFU notation throughout the manuscript, instead of 1792 x 104 it should be 1.792 x 104.
  2. 135 I think it's not about the viability of nods, but the number of CFU for RN
  3. 136-137 How could nodules reach their highest population size counted in CFU?!
  4. Table 1. The result for pallida VS CO 30 g/kg IT for RhR seems illogical? Why such a discrepancy only for this system? It is difficult to assess in thepeer- review process because the method of calculating the IT is not described in the methods.
  5. in the footer for Table 1 - no explanation of what value > 1 is meant
  6. It is completely unclear what the values ​​below the lower horizontal line refer to in all tables; it takes a long time to think about it; this should be corrected for the reader's convenience and to avoid misinterpretations
  7. 157-158 You should avoid discussing them in the results section. There is a separate chapter for that.
  8. 162-163 as above; I do not agree with this .
  9. The title of 2.3. does not reflect of the content („symbiont-associated microorganisms”?)
  10. Figure 1. Add standard deviations to graphs.
  11. 208 add a reference to Fig. 2A (if it is missing in other places, please add it as well, with such a large amount of numerical data the reader may get lost). Moreover, it is unclear to me where this 7.7% came from.
  12. 212 What did the authors mean by 'microbial symbiotic activity' and how does this correlate with the results?
  13. 327-328 The results presented by the authors are in contrast to the known literature and the mechanisms of action of xenobiotics on microorganisms. How to explain this?

The introduction and discussion are completely incomprehensible because:

  1. The English used by the authors is completely incomprehensible. In most cases, I had to either guess or consult the source literature. (l. 69-72, 82-85,
  2. The use of very long sentences (several lines) with many threads separated by dashes, colons, etc. additionally contributes to the lack of clarity of the statements.
  3. Sentences and paragraphs are written without any causal relationship.
  4. In introduction chapter the Authors did not included the descrition/definitione of hormesis. It is also hard to understand the meaning of „the indicators of toxicity” and index of hormesis (l.61-63). Moreover, according to my knowledge, hormesis is a positive effect of small doses of a factor that has a negative effect at higher doses and not as it was included in l. 63.

Discussion chapter in particular:

  1. Is too extensive, but the quantity doesn't match the quality. Example: l. 350-353 completely out of context; l. 429 - quoting such detailed parameters is completely unnecessary.
  2. Very long sentences with a lot of related information Example: l. 301-306
  3. I suggest marking which results concern current research, e.g. by using the phrases "in the present/our study" etc. It is difficult to understand in the presented way.
  4. Many conclusions drawn by the Authors are not supported by data obtained in the current study. They are just are just suspicions without any support from the results.

Examples: l. 298 ‘effects on root-rhizobia symbiosis” (it is difficult to say because the parameters of symbiosis, such as nitrogen fixation, were not measured); l. 309 „possible because the root …”; l. 313-314; l. 334-337, l. 347-349, l. 354-357, l. 359-361,

  1. Other conclusions are completely meaningless (l. 369-379, l. 432-) and the context presented indicates the use of AI in writing this chapter without a critical look at the final result. Also format: using the phrase: e.g. [45] states that something looks for AI.
  2. 296 what does the abbreviation CP used here and in 7 other places mean?
  3. 316, l. 319 factual error; jasmonic acid and other compounds mentioned are plant hormones, not polysaccharides!
  4. 340 ‘attraction of rhizobia’ to roots by their exudates is different process (chemotaxis) than induction of Nod production; thus such sentence can be ununderstood by readers
  5. 344 what does it mean ‘inhibition of nodules’ ? Their production or inhibition of activity of nodules; Authors should be more precise
  6. 345 why write about heavy metals; what does it have to do with the authors' results?
  7. 353 what does 'contains 63%' mean in this part; endophytic bacteria can be located in different plant tissues
  8. 358 The authors observed the process, not its induction
  9. 368 what is this statement referring to?
  10. 380-382 It's hard to figure out what these sentences are about; different parameter values ​​for the same events?
  11. 426 since the effect of CO on morphology was not studied using SEM, it is rather difficult to talk about the effect on structure
  12. From line 444 to the end I suggest the authors review it to remove similar defects as mentioned earlier sincey the examples provided are only representative and do not reflect the scale of the problem.

Material and methods

  1. 558 does 'proliferation' refer to nodules?
  2. 558-560 needs to be reworded because it is incomprehensible
  3. 570 since RhR refers to rhizospheric bacteria, it is unjustified to describe them as endophytic, they were not isolated from plant tissues
  4. 573 „soil from rhizosphere” and not „rhizospher”
  5. 574 What kind of medium?

The methods listed below must be described due to limited access to the source publications. They are not in OpenAcess format. L. 576 stanining, l. 583 IT, l. 588 the method of THPs extraction

Chap. 4.6. delete; Instead it in chapter 4.5., paragraph 1, specify what the method was used for, also taking into account 4.6.

Comments on the Quality of English Language

very low, incomprehensible

Author Response

The authors have made the corrections indicates by both reviewers. The document includes some corrections indicated by Reviewe 1 (text highlighted in yellow). We area also sending you two documents containing the responses to the corrections for reviewer. 

Sincerely 

María del Carmen Rivera Cruz

Reviewer 2 Report

Comments and Suggestions for Authors

Review

Plants (MDPI)

The rhizosphere of Crotalaria hosts bioindicator symbionts of toxicity and induces the phytoremediation of the total hydrocarbons and linear alkanes of petroleum contamination

by

Ana Guadalupe Ramírez-May et al.

  1. General comments

    The article deals with certain microbial toxicity effects in the context of phytoremediation of petroleum compounds. The topic is of general interest and deserves being published.

    But, on the other hand, some relevant information is missing. The toxic effects only concentrate to symbiotic microorganisms present in the rhizosphere. Furthermore, there also exist recent publications which deal with normalized microbial toxicity tests which may be very relevant in this context (see: https://doi.org/10.3390/pr8111349 and https://doi.org/10.1007/s00253-024-13286-0). At least the discussion should be broadened to cover this important field. There should also be mentioned some microbial toxicity data using these normalized test systems with petroleum compounds.

    2. Special comments

    The title of the article is very difficult to understand. It should be reformulated to make it much easier. A proposal: The use of rhizosphere microorganisms for the determination of microbial toxicity to certain petroleum compounds

    l. 19. CO as an abbreviation is unlucky as it is already means carbon monoxide. I would propose CRO.

    The index of toxicity (IT) which is mentioned very often in the text is not explained in a sufficient way. There is a short explanation in Para 4.4, but this is far too short.

    l. 23 It should be “linear alkanes”

    l. 30/31: It should be “used for phytoremediation”

    There are some very complicated sentences in the text. The whole text should be corrected by a native speaker.

    CFU is often mentioned in the text. It should be mentioned that it means “colony forming units”.

    Table 1. What does NN mean?

    Table 2. What does colonization mean and how is it measured?

    Fig. 2. What does crudo oil mean? I think it is crude oil.

    l. 222. What do you mean with microbiological association?

    Table 3. It should be “length of carbon chain” instead of “carbon number, C”.

    Fig. 3. Please explain the canonical correspondence in detail.

    I think the Materials and Methods part should be before the Results part.

    Fig. 5 is very nice and illustrative.

    I think the discussion part should be enriched by an inclusion of data from normalized OECD and ISO microbial toxicity tests (see above). This would make this paper much more interesting.

    All in all, the research topic is worth being published, but a major revision seems to be advantageous.

Comments on the Quality of English Language

The language should be corrected by an English native speakeer

Author Response

The authors have made the the corrections indicated by reviewer 2 (test highlighted in green). We are also sending you two documents containing the responses to the corrections.

Sincerely 

María del Carmen Rivera Cruz

Round 2

Reviewer 1 Report

Comments and Suggestions for Authors

At this point, I would like to remind authors that the goal of a reviewer's work is to identify substantive errors and methodological deficiencies, but also to highlight the readability of the work for the average reader. Third parties more easily grasp context that is unclear to those not involved in preparing the manuscript. From my own experience, I know that corrections resulting from such critical comments significantly improve the reception of the reviewed work. Therefore, I reiterate my request to make corrections in accordance with the guidelines (attached). What is obvious to authors will not necessarily be obvious to readers. Precise description and adherence to facts constitute good practice in preparing manuscripts. High-quality corrections and language style also contribute to a positive reception (including that of reviewers).

For example, the writing format used by [46] in their work, instead of Smith et al. [46] in their research on... is rather sloppy and could indicate the use of AI.

Comments on the Quality of English Language

Still needs improvements.

Author Response

The recommendations were implemented, and manuscrip was revised. The corrections are highlighted in yellow, and corrections made toaddress the overall revisions requested by Reviewer 1 and 2 are highlighted in blue. The document´s wording was modified.. Manuscript attached.

I appreciate the time dedicated to reviewing the manuscript, making comments,  and providing recommendations,  which were incorporated at the authors discretion.

Sincerely

Maria del Carmen Rivera Cruz

Reviewer 2 Report

Comments and Suggestions for Authors

The authors have not considered my comments. I do not know whether it happened accidentally. Therefore, I would recommend another major revision taking into account my comments.

Comments on the Quality of English Language

The language should be corrected by an English native speaker

Author Response

Attached the revised and corrected. Notes; Green indicates Reviewer1, blue indicates corrections made address overall corrections requested by the reviewers. 

I appreciate the time dedicated to reviewing the manuscript, making comments and recommendation, which were included at the authors discretion.

Sincerely

Maria del Carmen Rivera Cruz

Round 3

Reviewer 2 Report

Comments and Suggestions for Authors

see attached comments

Comments on the Quality of English Language

The English language is now fine.

Author Response

Attached is a letter informing you of the manuscript, primarily regarding the toxicity test. Please try to incorporate the recommended articles. 

María del Carmen Rivera Cruz

Round 4

Reviewer 2 Report

Comments and Suggestions for Authors

The article is now fine and ready for publication